# PICASO: PERMUTATION-INVARIANT CONTEXT COMPOSITION WITH STATE SPACE MODELS

**Tian Yu Liu**[*]
UCLA

**Alessandro Achille**
AWS AI Labs

**Matthew Trager**
AWS AI Labs

**Aditya Golatkar**
AWS AI Labs

**Luca Zancato**
AWS AI Labs

**Stefano Soatto**
AWS AI Labs

## ABSTRACT

Providing Large Language Models with relevant contextual knowledge at inference time has been shown to greatly improve the quality of their generations. This is often achieved by prepending informative passages of text, or 'contexts', retrieved from external knowledge bases to their input. However, processing additional contexts online incurs significant computation costs that scale with their length. State Space Models (SSMs) offer a promising solution by allowing a database of contexts to be mapped onto fixed-dimensional states from which to start the generation. A key challenge arises when attempting to leverage information present across multiple contexts, since there is no straightforward way to condition generation on multiple independent states in existing SSMs. To address this, we leverage a simple mathematical relation derived from SSM dynamics to compose multiple states into one that efficiently approximates the effect of concatenating raw context tokens. Since the temporal ordering of contexts can often be uninformative, we enforce permutation-invariance by efficiently averaging states obtained via our composition algorithm across all possible context orderings. We evaluate our resulting method on WikiText and MSMARCO in both zero-shot and fine-tuned settings, and show that we can match the strongest performing baseline while enjoying on average $5.4\times$ speedup.

## 1 INTRODUCTION

Incorporating new information in deep learning models has traditionally been a costly process, often requiring re-training or fine-tuning their weights on new data. Fortunately, Large Language Models (LLMs) provide a compelling alternative: These models can 'learn' to leverage new contextual information in real-time by simply prepending them as inputs, without having to modify their weights (Ram et al., 2023). This has motivated a powerful application known as Retrieval-Augmented Generation (RAG), where LLMs are deployed with the ability to retrieve and incorporate relevant sources of information, or 'contexts', from vast external knowledge bases when queried by users at inference time.

Despite being faster than the naive fine-tuning of model weights, this approach still incurs significant computational costs. Not only must the system process the user query and generate an answer, but it must also process the retrieved context, which in real-world settings can amount to thousands of tokens. This problem is exacerbated in Transformer-based models, as the inference cost of generating output tokens scales quadratically with the length of the extended input (see Figure 1).

In contrast, State Space Models (SSMs) offer a more efficient alternative. SSMs encode information from arbitrary-length input sequences into a fixed-size state vector, which can then be conditioned on to generate new tokens without revisiting the original input. This suggests a simple method to reduce the cost of incorporating new contexts at inference-time: Instead of retrieving from a database containing raw context tokens, we can create a "database of states" containing pre-computed state representations of contexts. At inference time, generation starts directly from the retrieved state,

---

[*]Work conducted during an internship at Amazon

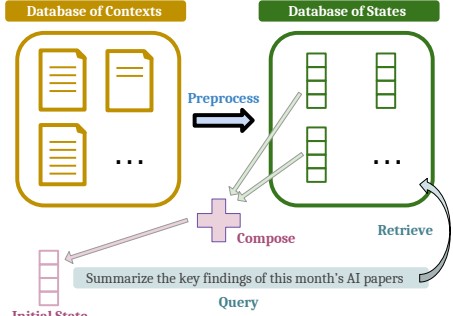 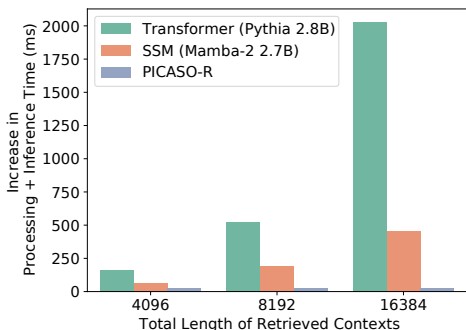

Figure 1: **(Left:)** We propose a "Database of States," where contexts are stored as pre-processed state vectors. Given a query, relevant states are then retrieved and composed into a single state vector which is used to condition the model's generation. **(Right:)** We plot the increase in total time required to generate an additional 64 tokens, when concatenating a 64-token prompt with retrieved contexts. We model the time taken for PICASO-R as the time taken to combine 5 pre-processed context states, which involves only arithmetic operations and notably zero model processing time. As a result, the processing and inference costs for PICASO-R remain constant regardless of the length of retrieved contexts. In contrast, the timings for a Transformer model scale quadratically, and for an SSM linearly, with total length when generating from concatenated context tokens. These timings are measured using the official Mamba benchmarking code, which includes optimizations such as quantization and CUDA graphs for SSMs, and flash attention for Transformers.

simultaneously eliminating the latency from having to process context tokens online, and greatly reducing inference time compared to Transformer models (Figure 1).

However, a key challenge arises when conditioning on multiple retrieved states. While input tokens can be simply concatenated and fed into an LLM or SSM, existing models are trained to generate outputs conditioned only on a single SSM state. To address this, we derive a simple mathematical relation via SSM dynamics to compose multiple states into one, in a manner that exactly equates to the result of concatenation in a single-layer model. Consequently, by simply storing an additional weight matrix along with each context, pre-computed states can be effectively composed at inference time to condition generation on any arbitrary set of contexts.

Since states are computed causally, the order in which contexts are presented affects the state – when there is no natural ordering among retrieved contexts, different order of presentation would yield different states. Consequently, we propose to enforce order-invariance explicitly through averaging states obtained by composing contexts across all possible permutations. While this may appear costly at first glance, we show that the resulting state can be computed exactly in polynomial time in the number of context segments using Dynamic Programming, and this can be further reduced to linear time by accepting slight approximations. This greatly benefits Retrieval-Augmented Generation tasks, where our results show a $10\%$ improvement over the best order-dependent state composition method when order-invariance is incorporated into the conditioned state.

To outline our main contributions, we introduce a method for efficiently retrieving and composing multiple pre-computed states at inference time to condition the generation of high-quality outputs, which we term PICASO (**P**ermutation-**I**nvariant **C**ompositional **A**ggregation of **S**tates as **O**bservations). Our experiments show that PICASO achieves 91% of the performance gain from combining the raw tokens of multiple contexts, while offering a $5.4\times$ speed-up over concatenation.

PICASO can be applied to any off-the-shelf SSM model without any changes. To further improve performance, we introduce a method for fine-tuning the model to better leverage the composed states for generation. Using a pre-trained Mamba-2 2.7B model, less than a day of fine-tuning on a single A100 GPU leads to the same performance as concatenation while maintaining the $5.4\times$ faster composition time on the WikiText-V2 dataset.

## 2 RELATED WORK

**State Space Models and Hybrid Models.** Recent efforts to overcome the significant computational costs of Transformer models on long contexts have inspired the exploration of more efficient alternatives, including State Space Models (SSMs). Through maintaining a fixed-size "state", a sufficient statistic of the past for the purpose of future prediction, these models offer advantages compared to Transformer models. They only require constant memory consumption regardless of the sequence length, and linear computational complexity, rather than quadratic, as longer sequences are processed. The idea of leveraging recurrent models with fixed dimensional states to represent complex sequences is not new, in fact, several variations of SSMs have been developed in the past, ranging from Linear Time Invariant (LTI) systems, to more expressive non-linear Time Varying (Jazwinski, 2007) and Input Varying (Krener, 1975) systems.

More recently, many of these ideas have been rediscovered and implemented on modern parallel hardware as basic building blocks for Foundation Models. Gu & Dao (2023) proposed Mamba, an input-dependent linear SSM (termed 'selective') based on LIV systems, that achieves comparable performance to Transformers (Vaswani, 2017) on language modeling while enjoying faster inference. Mamba-2 (Dao & Gu, 2024) further improved computational time by implementing SSM layers with structured matrix multiplications to better leverage modern Tensor Cores. Although pure SSM models can compete with Transformer blocks on many NLP tasks, they lag behind on tasks that require strong recall capabilities. To balance inference efficiency and model capabilities, Hybrid models combining Attention and SSM blocks have been proposed. Lieber et al. (2024) combined SSM blocks along with global-attention blocks to create a hybrid architecture with Mixture-of-Expert layers for training larger models. To further improve long context ability and efficiency, Ren et al. (2024) leveraged sliding window attention, while Zancato et al. (2024) developed a general family of architecture that include Transformers, SSMs and their hybrid combinations, leveraging both verbatim and fading memory, in both long- and short-term.

**Retrieval-Augmented Generation and In-Context Learning.** Our work falls within the scope of In-Context Retrieval-Augmented Language Models (Ram et al., 2023), where language models are conditioned on retrieved contexts via concatenation. Retrieval Augmented Generation (RAG) allows language models to leverage knowledge stored in external databases, which greatly improves performance on knowledge-intensive and domain-specific tasks (Lewis et al., 2020). In our work, we simply use a pre-trained sentence embedding model for retrieval, and we refer to Gao et al. (2023) for a detailed survey on other mechanisms. Apart from retrieval, processing (multiple) retrieved contexts can also greatly increase generation latency. Izacard et al. (2023) mitigates this by independently processes each retrieved context with a LLM encoder, using cross attention over the concatenated encoder outputs. Zhu et al. (2024) similarly encodes retrieved contexts in parallel, and performs decoding in a selective manner by attending only to highly relevant encoder outputs.

In-Context Learning (ICL) (Brown et al., 2020) has emerged as an effective method to perform inference without learning (*i.e.*, transduction), by conditioning on labeled samples provided in-context, commonly implemented as a set of (query, answer) pairs (Dong et al., 2022). Similar to RAG, the quality of selected demonstrations have been shown to greatly affect downstream performance (Xu et al., 2024). Several methods have been developed for selecting effective demonstrations, based on sentence embeddings (Liu et al., 2021), mutual information (Sorensen et al., 2022), perplexity (Gonen et al., 2022), and even BM25 (Robertson et al., 2009). Similar to the motivation of our work, several studies have shown that the performance of ICL is heavily dependent on demonstration ordering. Zhao et al. (2021) shows that answers positioned towards the end of the prompt are more likely to be predicted, while Lu et al. (2021) shows that results can vary wildly between random guess and state-of-the-art depending on the order that demonstrations are presented. Outside of ICL, Liu et al. (2024) further shows that language models do not robustly utilize information in long input contexts due to sensitivity to positioning.

**Model and State Composition.** Our work falls into the category of composing of deep models, representations, and states. Wortsman et al. (2022) proposes Model Soup, which composes multiple non-linearly fine-tuned models via averaging model weights. Liu & Soatto (2023); Liu et al. (2023) leverages model linearization to enforce an equivalence between weight averaging and output ensembling. Perera et al. (2023) independently learns task-specific prompts which can be linearly

averaged to yield new prompts for composite tasks. For SSMs, Pióro et al. (2024) investigates averaging of states, along with decay-weighted mixing which is closely related to a baseline version of our method, CASO. However, the equations described in their work differ from CASO, and their evaluations are limited to composition of two equal-length contexts. In contrast, our method greatly improves upon CASO by incorporating permutation invariance, which we show is important to achieve performances comparable to that of concatenation.

## 3 METHOD

### 3.1 PRELIMINARIES:

A linear input-dependent discrete-time state-space model has the form

$$\begin{cases} x_t = A(u_t)x_{t-1} + B(u_t)u_t \\ y_t = C(u_t)x_t + Du_t. \end{cases} \tag{1}$$

Here $x_t \in \mathbb{R}^m$ is the *state* at time $t$, while $u_t, y_t \in \mathbb{R}^d$ are the *input* and the *output* respectively. The matrices $A(u_t) \in \mathbb{R}^{m \times m}, B(u_t) \in \mathbb{R}^{m \times d}, C(u_t) \in \mathbb{R}^{d \times m}$ (which are input-dependent) and $D \in \mathbb{R}^{d \times d}$ are learnable parameters.

Unrolling the first equation, we obtain

$$x_t = A(u_t) \cdots A(u_1)x_0 + \sum_{\tau=0}^{t-1} A(u_t) \cdots A(u_{t-\tau+1})B(u_{t-\tau})u_{t-\tau}$$
$$= A(\boldsymbol{u})x_0 + x(\boldsymbol{u}), \tag{2}$$

where $\boldsymbol{u} = (u_1, \ldots, u_t)$ denotes the *sequence* of inputs, $A(\boldsymbol{u}) = A(u_t) \cdots A(u_1)$ is the accumulated decay matrix and $x(\boldsymbol{u}) = \sum_{\tau=0}^{t-1} A(u_t) \cdots A(u_{t-\tau+1})B(u_{t-\tau})u_{t-\tau}$ is the accumulated input signal. Since this coincides with $x_t$ when $x_0 = 0$, we refer to it as the *state* for input sequence $\boldsymbol{u}$.

In the following, we write $\mathcal{V} \subset \mathbb{R}^d$ for a finite set of token embeddings and $\mathcal{V}^* = \bigcup_{n \geq 0} \mathcal{V}^n$ for the set of variable-length sequences of token embeddings. We view a State Space (language) Model (SSM) as a map $f_\theta : \mathcal{V}^* \times \mathbb{R}^m \mapsto \mathbb{P}(\mathcal{V})$ with parameters $\theta$ which takes in as input an initial state $x \in \mathbb{R}^m$ and token embedding sequence $\boldsymbol{u} \in \mathcal{V}^*$, and returns a distribution over $\mathcal{V}$. Modern SSMs (Gu & Dao, 2023; Zancato et al., 2024) usually contain multiple stacked selective state space layers as in equation 1. In a multi-layer setting, we write $x(\boldsymbol{u})$ and $A(\boldsymbol{u})$ for the sequence of states and decay matrices corresponding to all layers.

### 3.2 DATABASE OF STATES

By the Markov property, the state of an SSM makes the past independent of the future. In other words, $f_\theta(\boldsymbol{u} \cdot \boldsymbol{u}', 0) = f_\theta(\boldsymbol{u}, x(\boldsymbol{u}'))$ for all $\boldsymbol{u}, \boldsymbol{u}' \in \mathcal{V}^*$, where $\cdot$ denotes concatenation. In practice, this means that a SSM model can equivalently be initialized with the state arising from a (variable-length) input sequence, instead of the input sequence itself. This is akin to the KV-cache of Transformer architectures, except that the dimension of the state is fixed regardless of sequence length.

In several real-world use cases such as Retrieval Augmented Generation, relevant contexts are commonly obtained or retrieved from a database (Borgeaud et al., 2022). Instead of storing them in the database as raw text or tokens, we propose to use a "database of states," where we pre-process each context and store their states. When conditioning on a single context, we can initialize the SSM with the retrieved pre-processed state instead of having to process it online. However this poses a problem when attempting to compose multiple contexts, since we do not know how to compose their states. We will show how this is tackled with our proposed method.

### 3.3 PERMUTATION-INVARIANT COMPOSITION WITH STATE SPACE MODELS

Given a query and a collection of relevant contexts, an easy method to compose them is to simply concatenate all context tokens with the query into a single sequence to feed into the SSM. Recall that this, however, presents two key limitations. Before even a single token continuation can be generated

from the query, the entire sequence of concatenated contexts has to be processed sequentially, which can be computationally intensive when contexts are long or numerous (Figure 1). Another limitation is having to select the order of context concatenation when prompting the model, for which there might be no natural way of doing so without a powerful scoring mechanism.

To address the first limitation, we propose a first version of our method, Compositional Aggregation of States as Observations (CASO), which works by modeling sequence concatenation with state composition based on the dynamics of a single-layer SSM.

**Proposition 1** (CASO). *Let $\boldsymbol{u}_1, \dots, \boldsymbol{u}_n$ be a collection of input sequences and let $\boldsymbol{u} = \boldsymbol{u}_1 \cdots \boldsymbol{u}_n$ be their concatenation. Then, for a SSM layer that evolves based on equation 1, we have*

$$x(\boldsymbol{u}) = x(\boldsymbol{u}_n) + \sum_{i=1}^{n-1} A(\boldsymbol{u}_n) \cdots A(\boldsymbol{u}_{i+1}) \cdot x(\boldsymbol{u}_i) \tag{3}$$

We can see this by recursively applying equation 2 on $x(\boldsymbol{u}) = A(\boldsymbol{u}_n) x(\boldsymbol{u}_1 \cdots \boldsymbol{u}_{n-1}) + x(\boldsymbol{u}_n)$.

Given a collection of contexts $\boldsymbol{u}_1, \dots, \boldsymbol{u}_n$, CASO simply approximates the dynamics of multi-layer SSMs, for which Proposition 1 does not hold exactly, via $x_\theta^{\mathrm{CASO}}(\boldsymbol{u}_1, \dots, \boldsymbol{u}_n) = x(\boldsymbol{u}_n) + \sum_{i=1}^{n-1} A(\boldsymbol{u}_n) \cdots A(\boldsymbol{u}_{i+1}) \cdot x(\boldsymbol{u}_i)$. We then load $x_\theta^{\mathrm{CASO}}(\boldsymbol{u}_1, \dots, \boldsymbol{u}_n)$ as the initial state of the model to infer continuations from the given query. We note that in Mamba-style models, the matrices $A(\cdot)$ are diagonal. As such, computing CASO requires only simple element-wise arithmetic operations and importantly zero model computation time (*i.e.* zero forward passes required).

However, since each state is weighted by the decay factors of future contexts, this composition operation is still very much order-dependent. We propose to introduce permutation-invariance by considering a group of permutations $G \subseteq S_n$, where $S_n$ denotes the symmetric group of $n$ elements, using which we define our method, PICASO (**P**ermutation-**I**nvariant CASO):

$$x^{\mathrm{PICASO}}(\boldsymbol{u}_1, \dots, \boldsymbol{u}_n) := \frac{1}{|G|} \sum_{\pi \in G} x^{\mathrm{CASO}}(\boldsymbol{u}_{\pi(1)}, \dots, \boldsymbol{u}_{\pi(n)}) \tag{4}$$

For any group $G$, by expansion of the CASO terms and collecting common factors, this can be written as a linear combination of individual context states $x(\boldsymbol{u}_i)$:

$$x^{\mathrm{PICASO}}(\boldsymbol{u}_1, \dots, \boldsymbol{u}_n) = \sum_{i=1}^{n} W_i(\boldsymbol{u}_1, \dots, \boldsymbol{u}_n) x(\boldsymbol{u}_i)$$

with weights $W_i$ depending on $A(\boldsymbol{u}_1), \dots, A(\boldsymbol{u}_n)$. In this work we are particularly concerned with two cases: the full symmetric group $G = S_n$, which includes all possible permutations, and the cyclic group $G = C_n$, which consists of rotations of the sequence. We will refer to them as PICASO-S and PICASO-R respectively.

While they appear computationally infeasible at first glance, since PICASO-S and PICASO-R average over $n!$ and $n$ CASO states respectively, each of which is itself a composition of $n$ context states, the following propositions show that they can actually be computed in polynomial and linear time respectively for modern SSM models with diagonal $A$ matrices.

**Proposition 2.** *Assume $G = S_n$ and that the matrices $A(\boldsymbol{u}_i)$ commute (e.g., are diagonal). Using shorthand notations $A_i := A(\boldsymbol{u}_i)$ and $W_k := W_k(\boldsymbol{u}_1, \dots, \boldsymbol{u}_n)$ we have*

$$W_k = \frac{1}{n!} \left[ (n-1)! + (n-2)! \cdot 1! \cdot \sum_{\substack{1 \leq i_1 \leq n \\ i_1 \neq k}} A_{i_1} + (n-3)! \cdot 2! \cdot \sum_{\substack{1 \leq i_1 < i_2 \leq n \\ i_1, i_2 \neq k}} A_{i_1} A_{i_2} + \dots \right]$$

$$= \frac{1}{n} \sum_{m=0}^{n-1} \frac{1}{\binom{n-1}{m}} \cdot e_m(A_1, \dots, A_{k-1}, A_{k+1}, \dots, A_n),$$

*where*

$$e_m(A_1, \cdots A_{n-1}) := \sum_{1 \leq i_1 < i_2 < \cdots < i_m \leq n-1} A_{i_1} \cdots A_{i_m}$$

*is the $m$-th elementary symmetric polynomial (Macdonald, 1998) (in the matrices $A_i$).*

Elementary symmetric polynomials satisfy the recursive relation

$$e_m(A_1, \ldots, A_{n-1}) = A_{n-1} e_{m-1}(A_1, \ldots, A_{n-2}) + e_m(A_1, \ldots, A_{n-2}).$$

Using this relation, we can compute all values of $e_m$, and hence the coefficients $W_k$, using $O(n^2)$ operations via Dynamic Programming. We detail the implementation in Algorithm 1 of the Appendix. Consequently, the full state from PICASO-S can be efficiently computed in polynomial $\mathcal{O}(n^3)$ time, which we show in the experiments to still be faster than processing textual context concatenations even for $n$ as large as 10.

Next, we similarly show that the coefficients for PICASO-R can be efficiently computed by exploiting invertibility of the matrices $A(\boldsymbol{u}_i)$.

**Proposition 3.** *Assume $G = C_n$ (cyclic permutations). Then writing $A_i := A(\boldsymbol{u}_i)$ and $W_k := W_k(\boldsymbol{u}_1, \ldots, \boldsymbol{u}_n)$ we have*

$$W_k = \frac{1}{n} \left[ \text{Id} + \sum_{m=1}^{n-1} A_{[k+m]_n} \cdots A_{[k+1]_n} \right].$$

*where* $\text{Id}$ *is the identity matrix, and $[i]_n$ denotes $i \mod n$. Assuming that the matrices $A_i$ are invertible, these can be computed efficiently by setting*

$$\bar{A}_i = \begin{cases} A_{[i]_n} \cdots A_1 \cdot A_n \cdots A_1 & i > n \\ A_i \cdots A_1 & i \leq n \end{cases}, \quad \bar{B}_i = \bar{A}_1 + \ldots + \bar{A}_{i-1}, \quad W_k = \frac{1}{n}[\bar{A}_k^{-1}(\bar{B}_{k+n} - \bar{B}_k)],$$

*for $i = 1, \ldots, 2n$, and $k = 1, \ldots, n$.*

We detail in Algorithm 3 in the Appendix our efficient implementation for computing PICASO-R in $\mathcal{O}(n)$ time complexity via cumulative sums and products. Evidently, PICASO-R is significantly faster than PICASO-S while trading off exact permutation invariance for invariance only to cyclic permutations of the original order. We will show that the difference in empirical performance between PICASO-S to PICASO-R is negligible, as such PICASO-S can almost always be replaced with its much faster variant PICASO-R.

We remark that the property of permutation-invariance can also be applied to naive concatenation (as opposed to CASO). This is achieved simply by concatenating contexts in various different orders, followed by taking an average of their resulting states. While performing this for the symmetric group $S_n$ is computationally infeasible, we can similarly restrict our permutation set to $C_n$. We term this variant Permutation-Invariant Concatenation (PIConcat-R), where $-R$ denotes invariance to the set of cyclic permutations. We note that the *model* computational costs (forward passes) of this method still scales quadratically with number of contexts (compared to linear scaling of regular concatenation), as such we include it only for completeness.

As a final technicality, we note that for Mamba-style SSM models, we additionally require storing the last $m_{conv}$ (usually $m_{conv} = 4$) input tokens of each SSM layer to ensure that the state is sufficient for generating the same distributions over continuations as the input sequence. We perform simple averaging to combine these tokens from different contexts which we show to work well empirically; more sophisticated methods could be explored in future work.

## 4 WHY PICASO'S AVERAGE WORKS

While the combination of state expression for CASO is directly motivated by the dynamics of the system, there is no a priori reason why averaging permuted CASO states should perform well. In Figure 3 we show that averaging both independent states and CASO states can perform better than using any individual state. This suggests a emergent/learned algebraic structure on the space of states such that linear combination of states combine their information to some degree.

In our empirical results below, we show that averaging all individual states (which would also be a permutation-invariant solution) performs significantly weaker than averaging CASO states (as PICASO does). We believe that this is because the approximate linear structure of the state space is only valid locally. The combined states are naturally closer together than the independent states, hence able to better exploit the local linearity. We show this in the following proposition:

**Proposition 4.** *Consider a single-layer SSM parametrized by $\theta$, and two input sequences $\boldsymbol{u}$ and $\boldsymbol{u}'$. Then, the Euclidean distance between the states can be bounded via*

$$\|x^{CASO}(\boldsymbol{u}, \boldsymbol{u}') - x^{CASO}(\boldsymbol{u}', \boldsymbol{u})\|_2^2 \leq \|(I - A(\boldsymbol{u}'))x(\boldsymbol{u})\|_2^2 + \|(I - A(\boldsymbol{u}))x(\boldsymbol{u}')\|_2^2$$

To see this, simply apply the triangle inequality on the following obtained via substituting the equations for CASO:

$$\|x^{CASO}(\boldsymbol{u}, \boldsymbol{u}') - x^{CASO}(\boldsymbol{u}', \boldsymbol{u})\|_2^2 = \|A(\boldsymbol{u}')x(\boldsymbol{u}) + x(\boldsymbol{u}') - (A(\boldsymbol{u})x(\boldsymbol{u}') + x(\boldsymbol{u}))\|_2^2$$
$$= \|(A(\boldsymbol{u}') - I)x(\boldsymbol{u}) + (I - A(\boldsymbol{u}))x(\boldsymbol{u}')\|_2^2$$

As a special case, we observe that as the decay factor approaches the identity, the distance between two CASO states approaches zero. In Figure 2, we visualize naive averaging of the states arising from 3 retrieved contexts, and averaging of CASO states resulting from each cyclic permutation of these contexts. We use WikiText-v2 as described in the Experiments for these plots. Indeed, we observe that CASO states are much closer to one another in the resulting loss landscape.

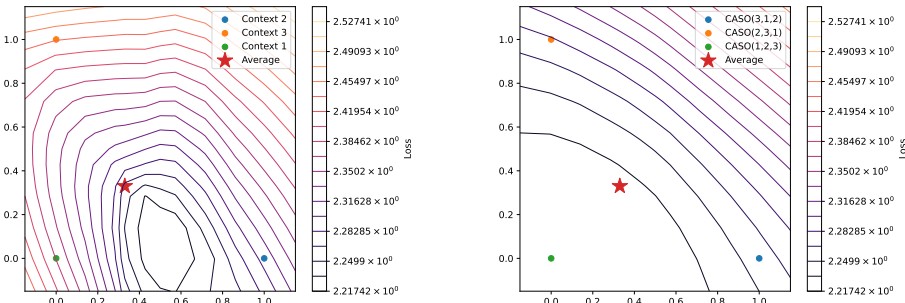

Figure 2: **Left:** Naive averaging ("Soup") of context states. **Right:** Averaging CASO states. CASO states are "closer" to one another (see Proposition) and hence can be more meaningfully interpolated. On the other hand, naively averaging states of independent contexts do not possess this property. Both plots are computed over 10 samples of (query, continuation, retrieved contexts).

## 5 LEARNING TO USE COMPOSED STATES

As previously noted, in practice, for SSM models consisting of multiple state space blocks stacked with temporal convolutions, $x(\boldsymbol{u})$ in equation 3 will not be exactly the state arising from a concatenated list of inputs. In this section, we introduce a fine-tuning objective to enable SSMs to better leverage composed states. Let $\mathcal{D} = \{(\boldsymbol{u}_i, u_i, S_i)\}_{i=1}^N$ be a dataset of sequences $\boldsymbol{u}_i$, their next-token continuation $u_i$, and a collection (in some particular order) of contexts $S_i$ retrieved from a database using $\boldsymbol{u}_i$. We minimize the prediction loss over the continuation, given a (composed) initial state and the query sequence:

$$\mathcal{L}_{BPTC}(\theta) = \sum_{(\boldsymbol{u}_i, u_i, S_i) \in \mathcal{D}} L_{\mathrm{CE}}(f_\theta(\boldsymbol{u}_i, x^{\mathrm{PICASO}}(S_i)), u_i),$$

where $L_{\mathrm{CE}}(\cdot, \cdot)$ is the cross-entropy loss.

We denote this learning objective Backpropagation Through Composition (BPTC), where gradients are propagated through the state composition process $x^{\mathrm{PICASO}}$. To reduce training time, we also consider an alternative version where we do not backpropagate through the composition step, which we denote Backpropagation To Composition (BP2C):

$$\mathcal{L}_{BP2C}(\theta) = \sum_{(\boldsymbol{u}_i, u_i, S_i) \in \mathcal{D}} L_{\mathrm{CE}}(f_\theta(\boldsymbol{u}_i, \mathrm{sg}\left[x^{\mathrm{PICASO}}(S_i)\right]), u_i),$$

where sg denotes the stop-gradient operator. We will show that when used for fine-tuning, this learning objective greatly improves the model's ability to leverage composed states for generation to the level of the concatenation albeit with much faster speeds, while maintaining performance on standard LLM evaluation tasks.

# 6 EXPERIMENTS

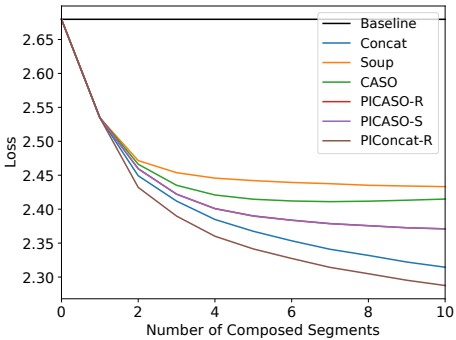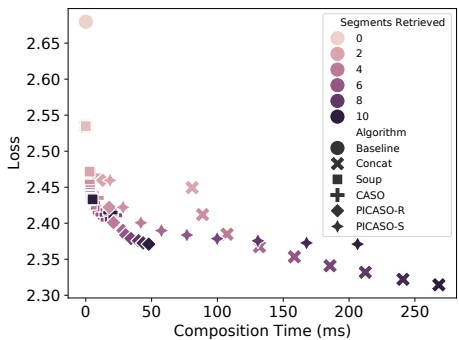

Figure 3: Zero-shot evaluation of PICASO using Mamba-2 compared to other composition methods on WikiText. While the performance of PICASO lags slightly behind that of concatenation (left), PICASO-R is on average $5.4\times$ faster (right). PICASO-S and PICASO-R perform similarly and yield overlapping curves (hence not visible in the left plot). Incorporating permutation invariance for concatenation via PIConcat-R gives the best results. However, it incurs magnitudes higher computational costs despite being performed within a single batched forward pass, hence we omit from the right plot to prevent it from disrupting the scale of the x-axis and focus comparisons on PICASO.

## 6.1 IMPLEMENTATION DETAILS

We run our main experiments on the largest available SSM on Huggingface - Mamba-2 2.7B (Dao & Gu, 2024). We evaluate our method on two large-scale datasets - WikiText-V2 (Merity et al., 2016) and MSMARCO (Nguyen et al., 2016). We use the training splits as our fine-tuning data, and the testing/validation splits respectively for evaluation. To pre-process WikiText-V2 for our use case, we split each passage in the dataset into two equal context "segments", with the goal of predicting the second (continuation) from the first (query). The retrieval database comprises all remaining segments, from which we retrieve via an external sentence embedding model, All-MiniLM-L6-v2[1]. In most experiments, we retrieve up to 10 segments, since improvements appears to saturate beyond that, and loss from concatenation blows up as a result of exceeding training context length (Figure 6, Appendix). We pre-process MSMARCO by filtering only entries with well-formed answers and discarding those without relevant passages.

We used the official benchmark[2] with an A100 GPU for our timing experiments in Figure 1 to ensure fairest comparisons. For the rest of the experiments, we run the model in full-precision, and evaluate performance of the model starting from a custom initial state, a feature not supported by the official benchmark at the time of writing, as such timings differ.

For fine-tuning experiments using BPTC and BP2C, we base our implementation on the official HuggingFace [3] trainer with default hyperparameters, and retrieve the $k$ most relevant context segments for each query sample for composition. For WikiText, we select $k \in \{0, \ldots, 10\}$ uniformly at random for each batch. For MSMARCO, we use all the available passages (both relevant and irrelevant) associated with each training example. For both datasets, we fine-tune for only 1 epoch. In all fine-tuning experiments, we ensure the training set (both the examples and the context database) are disjoint from the validation set to ensure fair evaluation.

## 6.2 COMPARISON MODELS

We compare inference accuracy (measured by log-perplexity) and processing latency of PICASO with its order-dependent version, CASO, in addition to the following methods:

---

[1]https://huggingface.co/sentence-transformers/all-MiniLM-L6-v2
[2]https://github.com/state-spaces/mamba
[3]https://github.com/huggingface/transformers

**Baseline**: Loss of the model on the test sample without using any contextual information.

**Concatenation** (Ram et al., 2023): We concatenate individual context segments based on a specific ordering. For WikiText-V2 experiments, we consider the "best-case ordering" as determined by the sentence embedding model where more relevant contexts are closer to the query (at the end). We initialize the model with the state of the earliest context segment in the concatenation, which we assume to be available via pre-processing, and recompute the composed state from only the remaining ones.

**Soup** (Pióro et al., 2024): Simple averaging of states obtained from each context.

### 6.3 MAIN RESULTS

In this section, we evaluate both the zero-shot and fine-tuned performance of PICASO in Section 6.3.1 and Section 6.3.2 respectively, and show in Section 6.3.3 that the fine-tuned model does not overfit to the composition task. We also include additional experiments showing that LLM capabilities are not impacted by fine-tuning in Appendix B.4, and show that PICASO can also be used for data attribution in Appendix C

#### 6.3.1 ZERO-SHOT PERFORMANCE

We demonstrate in Figure 3 that applying PICASO-R in a zero-shot manner on WikiText-V2 greatly improves performance over the baseline by an average of $10.1\%$ across 1-10 retrieved context segments. This greatly improves over Soup $(8.5\%)$ and CASO $(9.2\%)$. Compared to concatenation $(11.1\%)$, PICASO-R performs slightly worse but benefits from magnitudes improvement in processing time on an average of $5.4\times$. In this task, PICASO-R achieves almost exactly the same performance as PICASO-S, but with a much faster composition time. As a sanity check for motivation for our method, we show that PIConcat achieves the best performance $(12.0\%)$ overall, but at the cost of significantly greater computational time despite our batched-inference implementation.

In Row 1 of Table 1, we show that applying PICASO-R and PICASO-S in a zero-shot manner on MSMARCO similarly yields considerable improvements $(37.2\%)$ over the naive baseline, while achieving performance close to that of concatenation $(41.3\%)$.

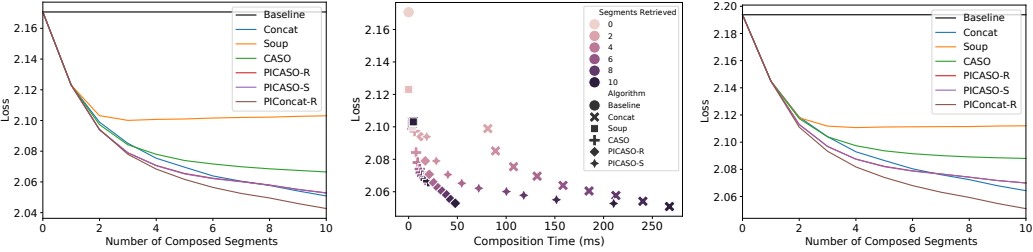

Figure 4: **(Left + Middle:)** Fine-tuning with BPTC on WikiText brings the performance of PICASO to that of concatenation, while retaining its significant speed advantages. **(Right:)** Fine-tuning with BP2C on WikiText improves the effectiveness of PICASO as well, but is much faster in terms of training time since it does not require backpropagating through the composed state. Note that fine-tuning has no impact on the actual composition time when used for inference.

#### 6.3.2 BACKPROPAGATION THROUGH AND TO COMPOSITION

While PICASO demonstrates strong performance in the zero-shot setting, PICASO still lags behind concatenation in terms of prediction accuracy. We attribute this to composed states being "out-of-distribution" for the model, since these states do not arise from any sequence of input tokens. In this section, we test if this can be resolved via fine-tuning with PICASO-R composed states via BPTC and BP2C. Indeed, as we show in Figure 4, BPTC and BP2C greatly improves the performance of PICASO-R and PICASO-S to that similar to concatenation, while maintaining much faster processing timings on WikiText. Similarly, we show in Rows 4-5 of Table 1 that fine-tuning on the MSMARCO training set also levels the performance of PICASO with that of concatenation. We also

Table 1: All models in this table are evaluated on the MSMARCO validation set. We evaluate performance of models fine-tuned via BPTC/BP2C on both the WikiText and MSMARCO training sets. Rows 2-3 show that fine-tuning models to compose WikiText context segments does not harm performance when evaluated on composing context segments from MSMARCO. When composing segments from distributions similar to those encountered during training (Rows 4-5), PICASO matches the performance of concatenation while being magnitudes faster.

|  | Naive | Concat | Soup | CASO | PICASO-R | PICASO-S |
|---|---|---|---|---|---|---|
| Mamba2-2.7B Base | 2.42 | 1.42 | 2.04 | 1.56 | 1.52 | 1.52 |
| Mamba2-2.7B BP2C-WikiText | 2.44 | 1.44 | 2.07 | 1.53 | 1.50 | 1.50 |
| Mamba2-2.7B BPTC-WikiText | 2.43 | 1.46 | 2.08 | 1.53 | 1.49 | 1.49 |
| Mamba2-2.7B BP2C-MSMARCO | 1.85 | 0.68 | 1.27 | 0.72 | 0.69 | 0.69 |
| Mamba2-2.7B BPTC-MSMARCO | 1.79 | 0.65 | 1.20 | 0.68 | 0.65 | 0.65 |

note that while BP2C is significantly faster in terms of training time, it incurs a small performance trade-off compared to BPTC for both datasets, keeping number of training iterations constant.

### 6.3.3 EVALUATION OF FINE-TUNED MODEL ON OTHER DIFFERENT TASKS

We showed that models fine-tuned on a specific downstream task (training set) using BPTC/BP2C perform strongly when composing samples drawn from a similar distribution (test set). We further show in Table 1 that models fine-tuned on one domain (WikiText) can demonstrate small performance gains (or at the very least, no performance loss) when composing samples via PICASO on another domain (MSMARCO). Finally, we show in Appendix B.4 that fine-tuning models with BP2C/BPTC maintain (and occasionally even improve) performance on general LLM evaluation tasks compared against the original model.

## 7 LIMITATIONS AND DISCUSSION

We have proposed a method, PICASO, that enables efficient retrieval and composition of contexts by pre-processing their individual states. Without any training, our approach can handle the composition of information contained in up to 10 context segments in a manner that is order-invariant. PICASO notably requires zero online model processing time, since generation can begin directly from the composed states. When models are further fine-tuned with our proposed learning objective, states composed using PICASO perform comparably to those produced from the concatenation of context tokens, while offering on average a $5.4\times$ faster composition time.

Nevertheless, our method does have some limitations. When applied in a zero-shot manner, PICASO still lags slightly behind concatenation in terms of prediction accuracy. PICASO is also currently limited to architectures based on SSM layers. We leave as future work extension of PICASO towards recently popularized attention-based hybrid models, which require more sophisticated methods of composing key-value caches. Lastly, we also leave as future work the exploration of parameter-efficient fine-tuning methods such as adapters, which can be used to augment the model at inference time to enable state composition while preserving the original model's behavior.

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

## A   ALGORITHMS: PICASO-S AND PICASO-R

We show in Algorithm 1 how PICASO-S is computed in polynomial time via a dynamic programming approach based on Algorithm 2. In Algorithm 3, we also show how PICASO-R can be computed with linear time complexity. Time complexity is measured as the number of arithmetic operations required as a function of number of context states.

---

**Algorithm 1** PICASO-S- $\mathcal{O}(n^3)$
---
**Require:** States $x = \{x_i\}_{i=0}^{n-1}$, Decays $A = \{A_i\}_{i=0}^{n-1}$
  **return** $\sum_{i=0}^{n-1}$ PICASO-S-DP$(A_{-i}) \cdot x_i$
$\qquad\qquad\qquad\qquad\qquad\qquad\qquad\qquad$ ▷ $A_{-i}$ denotes all elements of $A$ except $A_i$

---

**Algorithm 2** PICASO-S-DP - $\mathcal{O}(n^2)$
---
**Require:** Decays $A = \{A_i\}_{i=0}^{n-1}$
  DP[:,:] $\leftarrow$ zeros$(n, n)$
  DP[0,:] $\leftarrow 1$
  $w \leftarrow 0$
  **for** $i = 1, \ldots, n-1$ **do**
    **for** $j = i, \ldots, n-1$ **do**
      DP$[i][j] \leftarrow$ DP$[i][j-1] + A_{j-1} \cdot$ DP$[i-1][j-1]$
    **end for**
  **end for**
  **for** $i = 0, \ldots, n-1$ **do**
    $w \leftarrow w + \frac{1}{n \cdot \binom{n-1}{i}} \cdot$ DP$[i][n-1]$
  **end for**
  **return** $w$

---

**Algorithm 3** PICASO-R - $\mathcal{O}(n)$
---
**Require:** States $x = \{x_i\}_{i=0}^{n-1}$, Decays $A = \{A_i\}_{i=0}^{n-1}$
  $\hat{x} \leftarrow 0, \quad \hat{A} \leftarrow [A_1, \ldots, A_n, A_1, \ldots, A_n]$
  $\hat{A}_\Pi = \text{cumprod}(\hat{A})$
  $\hat{A}_{\Sigma\Pi} = \text{cumsum}(\hat{A}_\Pi)$
  **for** $i = 1, \ldots, n-1$ **do**
    $w_i \leftarrow \frac{1}{n} \cdot \left( \left( \hat{A}_{\Sigma\Pi}[n+i-1] - \hat{A}_{\Sigma\Pi}[i] \right) \hat{A}_\Pi[i]^{-1} + 1 \right)$
    $\hat{x} \leftarrow \hat{x} + w_i \cdot x_i$
  **end for**
  **return** $\hat{x}$

---

## B   FURTHER ANALYSIS

### B.1   COMPUTATIONAL COSTS OF PICONCAT

In Figure 5, we visualize the computational costs incurred by PIConcat, which we show to dominate that of other methods despite resulting in the best performance on the WikiText dataset.

### B.2   SCALING BEYOND EFFECTIVE CONTEXT LENGTH

In Figure 6, we show that as the total length of retrieved contexts scale beyond a certain threshold (effective context size of the model), the loss from concatenation blows up and rapidly increases beyond the no-retrieval baseline. On the other hand, performance of PICASO remains stronger than that of the baseline when composing 50 context segments.

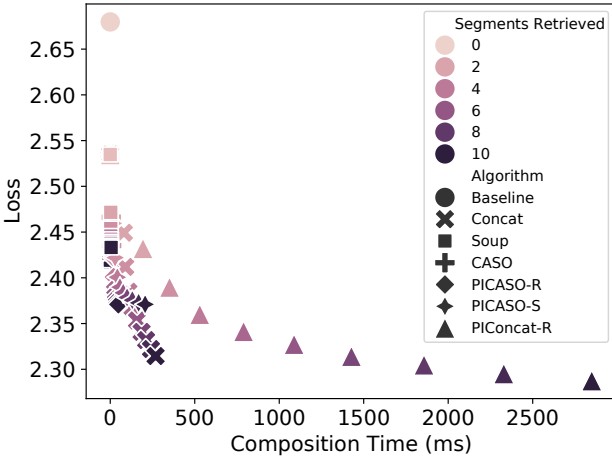

Figure 5: Timings for different composition algorithms evaluated on WikiText using Mamba-2 2.7B (zero-shot), including that of PIConcat-R. While PIConcat results in the best performance (y-axis), its computational cost (x-axis) is significantly higher than that of other methods. We refer to Figure 3 for a more condensed plot to compare the remaining methods.

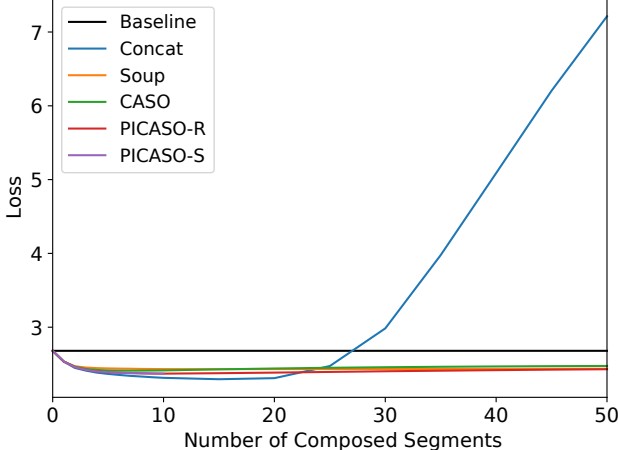

Figure 6: Concatenation scales poorly with total size of retrieved contexts beyond training context length. PICASO yields greater stability even composing up to 50 context segments retrieved from WikiText.

### B.3 INFERENCE VS PROCESSING TIME

In Figure 7, we show that the context processing time for Mamba-2 comprises a significant proportion of the total generation time. For large sequence lengths beyond 6K tokens, the processing time even dominates the inference time for generating 32 tokens.

### B.4 PERFORMANCE ON LLM EVALUATION TASKS

In Table 2, we show that fine-tuning Mamba2-2.7B with BTPC/BP2C objectives do not degrade existing LLM capabilities when evaluated on several LLM evaluation benchmarks - HellaSwag (Zellers et al., 2019), PIQA (Bisk et al., 2020), ARC-E, ARC-C (Clark et al., 2018), WinoGrande (Sakaguchi et al., 2021), and OpenbookQA (Mihaylov et al., 2018).

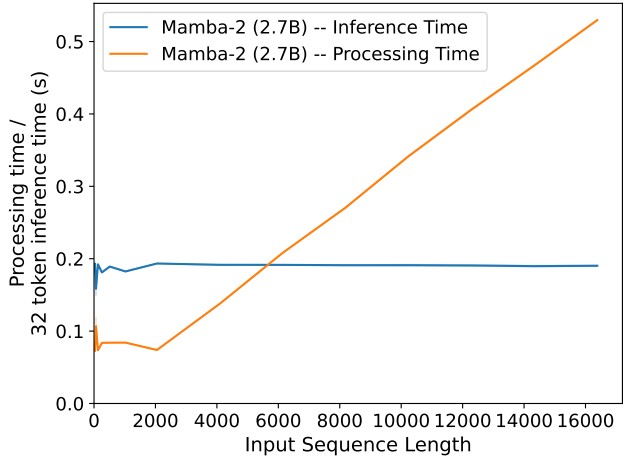

Figure 7: Mamba-2 Processing vs Inference Time of 32 tokens. Processing time (orange) occupies a significant proportion of the total time taken to generate from an input sequence, even dominating the constant inference time from the processed state (blue) as number of tokens in the input grows.

Table 2: Evaluation of Mamba2-2.7B trained with BPTC and BP2C on LLM evaluation tasks. Here, we show that fine-tuning for composition does not degrade existing LLM capabilities. In this table, we report the length-normalized accuracy for each task.

|  | HellaSwag | PIQA | ARC-E | ARC-C | WinoGrande | OpenbookQA |
|---|---|---|---|---|---|---|
| Mamba2-2.7B Base | $66.6 \pm 0.5$ | $76.3 \pm 1.0$ | $64.8 \pm 1.0$ | $36.3 \pm 1.4$ | $63.9 \pm 1.4$ | $38.8 \pm 2.2$ |
| BP2C-WikiText | $66.7 \pm 0.5$ | $76.3 \pm 1.0$ | $64.9 \pm 1.0$ | $37.5 \pm 1.4$ | $63.6 \pm 1.4$ | $39.8 \pm 2.2$ |
| BPTC-WikiText | $66.7 \pm 0.5$ | $75.6 \pm 1.0$ | $64.9 \pm 1.0$ | $37.2 \pm 1.4$ | $63.2 \pm 1.4$ | $40.2 \pm 2.2$ |

## B.5 ABLATION ON CHOICE OF RETRIEVER

In Figure 8, we ablate the impact of difference retriever choices on PICASO-R. In particular, we evaluate the performance of PICASO-R on WikiText when using the following embedding models from Sentence-Transformers (Reimers & Gurevych, 2019): average_word_embeddings_glove.6B.300d, all-MiniLM-L6-v2, and all-mpnet-base-v2, arranged in increasing order of performance on 14 different sentence embedding tasks (Reimers & Gurevych, 2019). As expected, Figure 8 shows that the performance of PICASO-R highly correlates with the strength of the retriever, where stronger retrievers yields better results on WikiText.

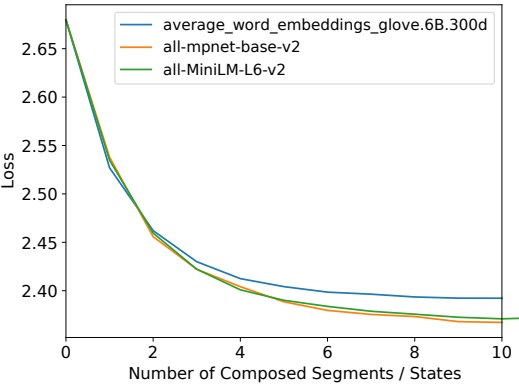

Figure 8: Ablation study on how choice of retriever model impacts performance of PICASO-R on WikiText. As expected, stronger retriever models result in better downstream performance.

### B.6 EVALUATION ON MULTIPLE CHOICE TASKS

In this section, we evaluate PICASO-R on the OpenbookQA (Mihaylov et al., 2018) multiple-choice task, where we retrieve from a context database of full passages from WikiText-V2. While OpenbookQA provides the ground truth fact for each evaluation sample, we discard this in our evaluations following standard practice in Gao et al. (2024). We leverage the same retrieval model used for the main WikiText experiments. Table 3 shows that PICASO-R achieves performance close to concatenation, with a $8\times$ speed-up in composition time.

| Naive | | Concat | | PICASO-R | |
|---|---|---|---|---|---|
| Acc ($\uparrow$) | Time ($\downarrow$) | Acc ($\uparrow$) | Time ($\downarrow$) | Acc ($\uparrow$) | Time ($\downarrow$) |
| 38.8% | NA | 40.0% | 233 ms | 39.9% | 29 ms |

Table 3: Evaluation on OpenbookQA dataset, augmented with retrieved passages from WikiText. We use normalized accuracy as our evaluation metric, and report the time taken to compose retrieved passages. Numbers shown in the table are averaged across retrieving between 1 to 10 full contexts from WikiText (as opposed to half context segments in our main paper experiments).

### B.7 CONTEXT STATISTICS

In Figure 9, we plot the distribution over the lengths (tokens and characters) of retrieved context segments used in the main paper WikiText retrieval dataset.

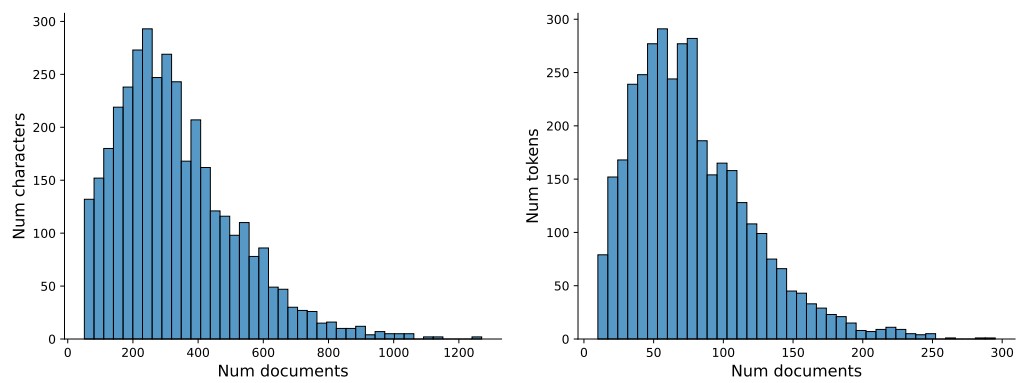

Figure 9: Histogram of the lengths, in terms of (**Left**) characters and (**Right**) tokens, of database context segments used in the main paper WikiText experiments.

## C DATA ATTRIBUTION

Table 4: Zero-shot Data Attribution on MSMARCO with Mamba2-2.7B, measured by precision. We compare Leave-One-In (LOI) and Leave-One-Out (LOO), where we implement LOO with varying methods for state composition.

| LOI | Concat | Soup | CASO | PICASO-R | PICASO-S |
|---|---|---|---|---|---|
| 0.699 | 0.690 | 0.629 | 0.725 | 0.732 | 0.731 |

Consider a question-answer pair $(\boldsymbol{u}_q, \boldsymbol{u}_a)$, and a sequence of potentially relevant contexts $S = (\boldsymbol{u}_1, \dots, \boldsymbol{u}_n)$. We would like to select the most relevant context for inferring the answer. There are at least two ways to do so with model $f_\theta$:

The first method, which we term "Leave-one-in", is to prepend each candidate context to the question, and evaluate the loss on the answer. Equivalently, $\arg\min_i L_{CE}(f_\theta(\boldsymbol{u}_q, x(\boldsymbol{u}_i)), \boldsymbol{u}_a)$, where we abuse notation to denote loss on the sequence (instead of token) $\boldsymbol{u}_a$.

The second method, which we term "Leave-one-out", is to compare the marginal increase in loss of the answer when removing each candidate from the composition of all of them. Equivalently, $\arg\max_i\{L_{CE}(f_\theta(\boldsymbol{u}_q, \hat{x}(S_{-i})), \boldsymbol{u}_a) - L_{CE}(f_\theta(\boldsymbol{u}_q, \hat{x}(S)), \boldsymbol{u}_a)\}$, where $\hat{x}(S_{-i})$ denotes a state composed from all contexts in $S$ other than $\boldsymbol{u}_i$.

Intuitively, the former measures "absolute" influence of a context, while the latter measures "relative" influence computed as the marginal improvement from adding it to the set of other contexts.

There are several different ways to implement the latter by varying the composition method used. We show in Table 4 that not only does Leave-One-Out perform best on the MSMARCO dataset, but implementing Leave-One-Out with PICASO-S and PICASO-R not only accelerates processing, but also surpasses the performance of conatenation. We attribute this to the permutation-invariance property of PICASO, which unlike concatenation, does not introduce irrelevant biases arising from arbitrary context orders.

## D  CONCATENATION FOR SSMS: CONNECTION TO JUMP-LINEAR SYSTEMS

Consider a collection of context segments retrieved based on relevance to a query, and sorted randomly as context to the query. While these segments share information, they are independent given the query, and their order is accidental and uninformative.

We are interested in a model that can efficiently process inputs in this format and extract all shared information from the input. Attention-based models are a natural choice because of the permutation-invariance of attention mechanisms (ignoring positional encoding), but they would have to process the entire input (all segments) with quadratic inference cost. On the other hand, SSMs have linear cost, but they are ill-fit to process this kind of input because of the context switches, which make the Markov assumption implicit in the state representation invalid.

We consider a broader class of models, namely switching dynamical systems (or jump-Markov, jump-diffusion, or linear hybrid, or jump-linear systems) as the class of interest. A jump-linear system is one that has a continuous state, say $x_t$ that evolves synchronously, and a discrete state that changes the value of $x_t$, for instance

$$x_{t+1} = \begin{cases} Ax_t + Bu_t & \text{if } t \in \mathcal{Z}\backslash\Omega \\ x_{t+1} \sim P & \text{if } t \in \Omega \end{cases}$$

Learning and inference for this model class corresponds to identification and filtering for this class of Jump-Markov models. In addition to a random switching, the switch can be triggered by a particular 'flag' (value) of the input:

$$x_{t+1} = \begin{cases} Ax_t + Bu_t & \text{if } u_t \neq u_{\text{trigger}} \\ x_{t+1} \sim P & \text{if } u_t = u_{\text{trigger}} \end{cases}$$

If the value of $u_{\text{trigger}}$ is known, then a given identification and filtering scheme can be applied by switching the estimated state according to the trigger.

Since modern state space models are input-dependent, they automatically fit the latter class of models and can handle switches without modifications. However, what they cannot handle is the fact that the order of the segments is uninformative. As a result, presenting the same segments in different order would yield different states. Accordingly, our goal is to enable SSMs to learn from segments up to permutations, so we can accommodate sequences where the ordering within segments is informative and respected, while the ordering of segments is uninformative and factored out.

## E  GENERAL RECURRENCE STRUCTURE

In the main paper, we introduced a specific recursive relation satisfied by Elementary Symmetric Polynomials. Here, we introduce a more general form which can potentially be used for more efficient implementations:

**Proposition 5.** *For any choice of* $1 \leq q \leq n - 1$

$$e_m(A_1, \cdots, A_{n-1}) = \sum_{j=max(q+m-n+1,0)}^{min(m,q)} e_{m-j}(A_1, \cdots, A_{n-1-q}) e_j(A_{n-q}, \cdots A_{n-1})$$

*Proof.* We compute $e_m(A_1, \cdots, A_{n-1})$ using a Dynamic Programming (DP) approach, where we break the problem into smaller problems, and merge the solutions. First we split the $n - 1$ variables at some random index $q$ to create two partitions, $(A_1 \cdots, A_{n-1-q})$ and $(A_{n-q}, \cdots A_{n-1})$, and then compute $e_{m-j}$ and $e_j$ on each partition respectively. For a given value of $j$, $e_{m-j}(A_1, \cdots, A_{n-1-q}) e_j(A_{n-q}, \cdots A_{n-1})$ will only compute a subset of values from $e_m(A_1, \cdots, A_{n-1})$, and hence we sum over all possible values for $j$. $\square$

In particular, taking $q = 1$, we obtain the following:

$$e_m(A_1, \ldots, A_{n-1}) = A_{n-1} e_{m-1}(A_1, \ldots, A_{n-2}) + e_m(A_1, \ldots, A_{n-2})$$

which we use for our implementation of PICASO-S.

