# OpenReview forum: "PICASO: Permutation-Invariant Context Composition with State Space Models"
_ICLR.cc/2025/Conference — ICLR 2025 Poster_

### Official Review · Reviewer_h4ZY · 2024-11-03

**Soundness:** 2
**Presentation:** 2
**Contribution:** 3
**Rating:** 5
**Confidence:** 4

**Summary:**

This paper proposes a way to speed up inference for RAG use cases for state space models (SSMs). The speed up comes from feeding in states of the retrieved chunks instead of doing inference from raw tokens to best utilize the properties of SSMs. To address the problem of combinatorial number of possibility in concatenation the authors proposed two ways of computing all the permutations -- exhaustive and cyclic. The authors demonstrate the proposed approach on two datasets, which shows improvements over naive concatenation of states. The performance compared to baseline concatenation is still worse but faster, the gap closes after finetuning.

**Strengths:**

The proposed method shows significant speed up over the naive baseline

**Weaknesses:**

The writing can be improved, in particular in the experiment section. It is not clear what the size of the chunk, what is the inference token size (how many in context and how many for inference), and how the evaluation is setup. For wikitext, it is not clear what is used as the query and what is used as the candidate pool for performing retrieval.

The evaluation feels incomplete. It seems that for both WikiText and msmarco are evaluated based on the test set loss. Since the proposed method fits in the RAG setting, it would be great to show the actual QA performance instead of the cross entropy as the result.

It is not clear what the speed baseline setup is. Does it use KV cache?

**Questions:**

See questions in the above section

---

> ### Comment · Reviewer_2fh2 · 2024-11-13
> **This reviewer is not reading paper carefully.**
>
> Putting other things aside, it's really interesting to see you asking about KV cache in an SSM paper (even if the state is analogous to KV cache). If you haven't read the paper, please lower your confidence.

---

> > ### Comment · Reviewer_h4ZY · 2024-11-13
> > **Meant for the transformer baseline comparison**
> >
> > Thanks for your comment. I think the comment is not clear.
> >
> > The authors did compare with a transformer baseline, and the kv cache question is regarding that transformer baseline number.

---

> > > ### Comment · Reviewer_2fh2 · 2024-11-13
> > >
> > > Good defense. But if it does not use KV cache, it would scale in O(n^3). Anyway, my point is clear. I do not think you read the paper carefully.

---

> > > > ### Author Response · Authors · 2024-11-13
> > > > **Quick Clarification**
> > > >
> > > > While we are currently preparing our official rebuttal, we would like to promptly address this concern in view of the increased attention it has received. Indeed, we tried to make our timing comparisons with transformers as fair as possible. We refer to L75-76, where we state
> > > >
> > > > > These timings are measured using the official Mamba benchmarking code, which includes optimizations such as quantization and CUDA graphs for SSMs, and flash attention for Transformers
> > > >
> > > > This naturally includes using the KV cache for generation, and we will update the relevant lines to make this explicit.

---

> > > > > ### Comment · Reviewer_h4ZY · 2024-11-13
> > > > >
> > > > > Thanks for the response.
> > > > >
> > > > > Maybe kv cache is the wrong word. What i meant is that since this method retrieves pre-computed states, one can compare with a similar transformer baseline, where the kv caches for documents are pre-computed and then can be directly loaded into the model for generation. If the original time already follows this setup then this question is addressed.

---

> > > > > > ### Author Response · Authors · 2024-11-19
> > > > > >
> > > > > > We thank the reviewer for their constructive feedback.
> > > > > >
> > > > > > > The writing can be improved, in particular in the experiment section. It is not clear what the size of the chunk, what is the inference token size (how many in context and how many for inference), and how the evaluation is setup. For wikitext, it is not clear what is used as the query and what is used as the candidate pool for performing retrieval.
> > > > > >
> > > > > > To address the reviewer's concern, we have added statistics on the document chunks used in our WikiText experiments in Appendix B.7, which should help clarify these implementation details regarding sizes of retrieval and query chunks. Our evaluation setup is detailed in Sec 6.1. -- the goal is to predict the second half of a document from its first, by leveraging retrieved knowledge from other document chunks.
> > > > > >
> > > > > >
> > > > > > > The evaluation feels incomplete. It seems that for both WikiText and msmarco are evaluated based on the test set loss. Since the proposed method fits in the RAG setting, it would be great to show the actual QA performance instead of the cross entropy as the result.
> > > > > >
> > > > > > We thank the reviewer for their suggestion. To address their concern regarding more evaluation methods, we have added section B.6. in the Appendix where we evaluate accuracy on a multiple-choice task. We observe the same trends (benefits of augmented generation, and strong performance of PICASO), as that when using loss/perplexity as our evaluation metric.
> > > > > >
> > > > > > > It is not clear what the speed baseline setup is. Does it use KV cache?
> > > > > > > Since this method retrieves pre-computed states, one can compare with a similar transformer baseline, where the kv caches for documents are pre-computed and then can be directly loaded into the model for generation.
> > > > > >
> > > > > > Fig 1 measures the necessary pre-processing time, which includes the creation of the KV cache for a transformer or the creation/composition of the state for SSM, along with the inference time starting from the processed cache/state.
> > > > > >
> > > > > > We highlight that pre-computation of the KV-cache does not work here, since KV caches cannot be composed (concatenating them is not valid). On the other hand, we can pre-process these states for PICASO, since our method provides a way to compose them.

---

> > > > > > > ### Comment · Reviewer_h4ZY · 2024-11-23
> > > > > > >
> > > > > > > Thanks fore the response, I have updated my score accordingly

---

> > > > > > > > ### Author Response · Authors · 2024-11-24
> > > > > > > >
> > > > > > > > Thank you for reviewing our response. We would greatly appreciate it if you could share any remaining feedback or suggestions. Otherwise if we have satisfactorily addressed your concerns, we hope that you would consider increasing your score to support our paper's acceptance.

---

### Official Review · Reviewer_2fh2 · 2024-11-03

**Soundness:** 4
**Presentation:** 3
**Contribution:** 4
**Rating:** 8
**Confidence:** 4

**Summary:**

The paper addresses the challenge of efficiently incorporating multiple documents into the generation process of LLMs. Traditionally, concatenating documents leads to significant computational costs that scale with the number and size of document chunks. State Space Models (SSMs) offer a faster approach by encoding documents into fixed-size state vectors, but composing multiple states is not straightforward. The authors introduce PICASO, a method for permutation-invariant composition of document states using SSMs. PICASO efficiently retrieves and combines pre-computed states to condition the generation of high-quality outputs without the need for online document processing. It enforces invariance to the order of document presentation, which is desirable when the temporal ordering is uninformative.

**Strengths:**

1. The issue is of significant importance. Generating content from multiple processed documents without re-preprocessing is a crucial requirement for both long context application and agentic memory systems.
2. The State Space Model constitutes a suitable architectural framework for the targeted problem, thereby rendering the study meaningful.
3. The introduced methodologies are technically robust. And they are presented clearly mostly (with a few exceptions).
3. The experiments are meticulously designed. Both performance and time complexity are analyzed. I particularly appreciate the experiments depicted in Figure 2, which elucidate the improvement brought about by the proposed method and provide mechanistic insights for the field.

**Weaknesses:**

I will write the weaknesses in the order of the willingness of raising my score after addressing these weaknesses.

1. Figure 3,4 (left) has an error. There are seven legends and only six curves. The main approach PICASO-R is missing from the figure. Though I infer from the context of different places (e.g., PIConcat can not run for 10 chunks) that it is a mistake that PIConcat-R's curve should actually be PICASO-R? But such a mistake seriously lowers the quality of the paper.

2. The differences between BPTC and BP2C are very ambiguous (And I actually do not know the difference). Equations are prefered. And stop-gradient notation can be used if you need to clarify the difference.

3. Clarify usefulness outside of Mamba-1. The PICASO-S relies on the commutativity of A_i, and the PICASO-R relies on the invertible property of A_i. Despite that Mamba-1 clearly holds these properties. It is not very clear whether other SSMs (e.g., RWKV, RetNet, Mamba-2, etc.) hold these properties. This is not a very important limitation of this paper, since work can also be done for other architectures, but it would be interesting to see the authors' analysis in this paper.

4. It seems that CASO is also the original proposal of the paper. Since I am not very clear about these baselines (But I do know RAG, Mamba, etc., clearly). I think it would be beneficial to mention methods similar to it more.

**Questions:**

See weakness.

---

> ### Author Response · Authors · 2024-11-19
>
> We thank the reviewer for their detailed feedback and constructive suggestions, which we have incorporated in our revision.
>
> > Figure 3,4 (left) has an error. There are seven legends and only six curves. The main approach PICASO-R is missing from the figure. Though I infer from the context of different places (e.g., PIConcat can not run for 10 chunks) that it is a mistake that PIConcat-R's curve should actually be PICASO-R? But such a mistake seriously lowers the quality of the paper.
>
> There is no mistake in the figure, but the curves of PICASO-R and PICASO-S actually overlap and are hence hard to see. We have made this explicit in the revised caption, and we thank the reviewer for noticing the readability issue.
>
> > The differences between BPTC and BP2C are very ambiguous (And I actually do not know the difference). Equations are prefered. And stop-gradient notation can be used if you need to clarify the difference.
>
> We thank the reviewer for their suggestion and notation advice, and have updated our revision with the equations in Sec. 5 to provide better clarity.
>
> We copy the definitions below, where we replace $\boldsymbol{u}_i$ with $v_i$ due to issues with rendering in Openreview
>
> $$\mathcal L_{BPTC}(\theta) = \sum_{(v_i,u_i,S_i) \in \mathcal{D}}L_{\rm CE}(f_\theta(v_i,  x^{PICASO}(S_i)), u_i)$$
>
> and
>
> $$\mathcal L_{BP2C}(\theta) = \sum_{(v_i,u_i,S_i) \in \mathcal D} L_{\rm CE}(f_\theta(v_i,  \operatorname{sg}\left[x^{\rm PICASO}(S_i)\right]), u_i)$$, where $\operatorname{sg}$ is the stopgradient operator.
>
>
>
> > Clarify usefulness outside of Mamba-1. The PICASO-S relies on the commutativity of A_i, and the PICASO-R relies on the invertible property of A_i. Despite that Mamba-1 clearly holds these properties. It is not very clear whether other SSMs (e.g., RWKV, RetNet, Mamba-2, etc.) hold these properties. This is not a very important limitation of this paper, since work can also be done for other architectures, but it would be interesting to see the authors' analysis in this paper.
>
> Our method works for both Mamba-1 and Mamba-2 since as rightfully pointed out by the reviewer, the $A_i$ matrices are diagonal hence commutative/invertible. Our experiments are performed using Mamba-2 2.7B. These properties are often satisfied by several recurrent mechanisms in order to make them computationally efficient, including RWKV (via element-wise scaling) and RetNet (where $A$ is parameterized as a diagonalized matrix).
>
> > 4. It seems that CASO is also the original proposal of the paper. Since I am not very clear about these baselines (But I do know RAG, Mamba, etc., clearly). I think it would be beneficial to mention methods similar to it more.
>
> Thank you for the suggestion, we added links to relevant literature when introducing our baselines in Sec 6.2. which we hope addresses the reviewer's concern.

---

> ### Author Response · Authors · 2024-11-30
>
> Given the approaching rebuttal deadline, we wish to follow up on our response to your insightful feedback. We hope that our responses, revised paper draft, and additional experiments have addressed the areas of weakness you identified. We would be grateful if you could consider providing stronger support for our paper's acceptance, and thank you for your very valuable feedback and suggestions!

---

> > ### Comment · Reviewer_2fh2 · 2024-12-03
> >
> > Thanks for your response. I take a close look at the paper and find that there is still one concern that BP2C still lags behind BPTC by about 0.01 in loss. But I think my previous concern has been addressed. I think the direction of this paper is worth exploring in the future need of inference scaling. Therefore I decide to encourage the authors by raising my score.

---

> > > ### Author Response · Authors · 2024-12-03
> > >
> > > Thank you for your support, and for updating your score. We sincerely appreciate your time and valuable feedback in reviewing our work!

---

### Official Review · Reviewer_FYyF · 2024-11-05

**Soundness:** 3
**Presentation:** 3
**Contribution:** 3
**Rating:** 6
**Confidence:** 1

**Summary:**

The paper presents PICASO, a method to enhance generation capabilities in Large Language Models by composing document states in a permutation-invariant manner using State Space Models (SSMs). This approach addresses the inefficiency and high computational cost of concatenating multiple document tokens by pre-processing document chunks into states. PICASO leverages a permutation-invariant composition strategy that enables LLMs to utilize multiple documents' information efficiently, achieving computational speed-ups and scalability, especially in retrieval-augmented tasks. Evaluation on datasets such as WikiText and MSMARCO shows that PICASO can match or closely approximate the performance of document concatenation with significantly reduced processing time.

**Strengths:**

1. PICASO offers an impressive 5.4x speed-up in processing time over traditional concatenation, a practical advantage for real-world applications that require high-speed document retrieval and composition.
2. The model effectively maintains performance without relying on the order of documents, which is crucial when temporal or logical ordering is irrelevant. This represents a thoughtful design that accommodates various real-world scenarios.
3. Extensive experiments demonstrate PICASO’s effectiveness, comparing multiple composition methods and showing zero-shot and fine-tuned settings. The authors provide a comprehensive view of PICASO’s strengths and limitations across several benchmarks.

**Weaknesses:**

1. While PICASO achieves near-concatenation performance in zero-shot scenarios, there is still a minor but noticeable gap in accuracy. This limitation could impact its suitability for applications where slight accuracy improvements are critical.
2. Although PICASO performs well with up to 10 document chunks, it is unclear how it scales with larger document sets. This potential bottleneck is worth further investigation.

**Questions:**

N/A

---

> ### Author Response · Authors · 2024-11-19
>
> We thank the reviewer for their feedback, and address points of concern below.
>
> > While PICASO achieves near-concatenation performance in zero-shot scenarios, there is still a minor but noticeable gap in accuracy. This limitation could impact its suitability for applications where slight accuracy improvements are critical.
>
> Our method is indeed targeted towards deployment of LLMs, where slight performance trade-offs are well justified by the substantial speed-up obtained. However, we note that trade-off can be easily avoided via our proposed fine-tuning method, obtaining the "best-of-both-worlds". Indeed in situations where fine-tuning is not an option and inference-time is a non-issue, concatenation remains the paragon.
>
> > Although PICASO performs well with up to 10 document chunks, it is unclear how it scales with larger document sets. This potential bottleneck is worth further investigation.
>
> This is a good point, and we have addressed this in Figure 6 of our main paper. While concatenation stops working as number of documents increase, due to the model becoming unstable once context length is exceeded, the performance of PICASO remains relatively stable even when composing up to 50 document chunks.

---

> > ### Author Response · Authors · 2024-11-30
> >
> > Since the rebuttal deadline is quickly approaching, we wish to follow up on our response to your review. We would be very grateful if you could share any further insights or suggestions you may have, otherwise if you feel we have satisfactorily addressed your concerns, we kindly ask if you could lend stronger support for our paper's acceptance. Thank you for your time and consideration!

---

### Official Review · Reviewer_uYJU · 2024-11-10

**Soundness:** 2
**Presentation:** 3
**Contribution:** 3
**Rating:** 5
**Confidence:** 4

**Summary:**

The paper focuses on the inference efficiency with state space models (SSMs) when conditioning on multiple document chunks in retrieval augmentation setups. It proposes a method to compose hidden states of different contexts into a single state for SSM generation, which is similar to compressing KV cache in Transformers (but for SSMs inference is only based on a single hidden state for one conditioning context). In particular, for applications that require retrieval, multiple document chunks could be provided as additional contexts, limiting the inference efficiency. Instead of the standard way of concatenating multiple document chunks in the context and run SSMs, the paper proposes to 1) preprocess individual chunks to store their SSM states; 2) retrieval the document chunks (with an additional retrieval model) and their pre-computed states based on a query; 3) compute an aggregated state from the different chunk states (which is not by simple averaging); and 4) continue generation based on the single aggregated state representing multiple document chunks.

The core contribution lies in step 3), where the authors derive an equivalent form (based on a single layer of SSM, so in general it is a heuristic) of state composition based on SSM computations to that of computed by naively concatenating multiple document chunks. Since this composition is reliant on a particular document chunk order, the paper further proposes to aggregate the composed states from different orders, assuming the order does not matter for the downstream generation. Efficient algorithms for the permutation-invariant aggregation are derived. Fine-tuning SSM by incorporating the aggregated states is also applied to further enhance the inference quality.

Experiments are conducted on language modeling tasks with the Mamba-2 2.7B model on two datasets, WikiText-V2 and MAMARCO. Results show that the proposed approach can achieve comparable perplexity to the baseline of directly concatenating multiple document chunks, while enjoying on average 5.4X speedup.

**Strengths:**

- The paper presentation is mostly clear. The motivation, background, and techniques are well explained, also in good connection with relevant research.

- The paper studies an interesting problem, which is to speed up inference in the context of retrieval augmentation, specifically focusing on SSM models with their unique state dependencies. SSMs do not need KV cache like Transformers thus reducing memory and computation requirements. The paper further improves inference efficiency by delegating the document state computation to the preprocessing time, and only compute state compositions for inference.

- The technicality is sound (I did not check the math in every detail but overall they seem correct) and interesting. Algorithms of efficient computation of document state composition are derived, as well as for fine-tuning SSMs with the state composition that utilizes SSM model parameters.

- Experimental studies support the expected results of maintaining language modeling performance, measured in log-perplexity, while speeding up the generation from the composite SSM states from multiple document chunks.

**Weaknesses:**

1. While the problem and methodology is interesting, I find the experimental studies are somewhat not adequate.
- The authors conduct experiments with simplified settings of retrieving from WikiText and MSMARCO document chunks to improve next token prediction perplexity. This is similar to in-context retrieval [1] (seems a missing reference). However, there are no real RAG applications studied in the paper. Appendix B.4 presents some additional results on “language modeling” (which I don’t think is a proper description), but not so much details are provided such as database and retrieval.
- The evaluation metric mostly focuses on log-perplexity or loss, while it is also desirable to demonstrate generation capabilities of the proposed method for real tasks.
- Furthermore, there could be certain baselines missing, including similar approaches proposed in the previous literature, such as the closely related work described in Section 2.
- Comprehensive ablation studies are missing, making it difficult to understand the experimental details such as what parameters in multi-layer SSMs are used and how that matters. Other ablations could include the effect of retrieval accuracy, the impact of document or chunk lengths to be composed together, etc.

2. Not enough background information is provided for RAG applications with SSMs, as most of the RAG applications are built with Transformers. This is also related to the lack of experimental studies mentioned above, where more comprehensive comparisons on other tasks/methods could be beneficial.

3. The method is based on some strong assumptions such as the order of the retrieved document chunks does not matter, so that an average state can be used for inference. There is no clear evidence provided.

4. Some of the descriptions are unclear. For example, some math notations are undefined, some details in Figure illustrations are missing. See more below in my questions.

5. It seems the gain of the proposed approach mostly comes from the inference efficiency side, as the log-perplexity (demonstrated in Table 1) is not improved over the baseline of document concatenation. However, the complete pipeline includes preprocessing document chunks for their SSM states (Figure 7 also shows this), and it also includes an additional retrieval model with a separate set of embeddings that need to be stored in the database. The data used for experiments are small. The considered number of chunks are smaller than 10. It is unclear how the proposed method performs when brought to larger scales requiring bigger storage and more computation in real applications.

*[1] Ori Ram, Yoav Levine, Itay Dalmedigos, Dor Muhlgay, Amnon Shashua, Kevin Leyton-Brown, and Yoav Shoham. 2023. In-Context Retrieval-Augmented Language Models. Transactions of the Association for Computational Linguistics, 11:1316–1331.*

Overall I think the paper presents an interesting method and study on Mamba states composition, but the empirical justifications are somewhat flawed. I am willing to increase my score after getting more insights from the authors regarding my concerns.

**Questions:**

1. In line 084-085, “relative ordering is often uninformative”: this is a very strong assumption. Any evidence? Especially in applications where retrieval meets Mamba. In fact, many previous studies show the position of information matters [2].

*[2] Nelson F. Liu, Kevin Lin, John Hewitt, Ashwin Paranjape, Michele Bevilacqua, Fabio Petroni, and Percy Liang. 2024. Lost in the Middle: How Language Models Use Long Contexts. Transactions of the Association for Computational Linguistics, 12:157–173.*

2. For Mamba models of multiple layers, what hidden states and which layer parameters did you use such as the A and B matrices? Is there any difference depending on what you use?

3. In Figure 2, can you explain more on how to read the figure? In particular, how does the figure show that the CASO states are closer to one another?

4. For experiments on val/test sets, what are the chunk databases? Do they include chunks in the training data, or just the chunks of corresponding val/test sets?

5. Since you use a different retrieval model, I assume you have to store the chunk vectors based on the retrieval model as well, on top of the SSM hidden states. Is that right? What is the effect on retrieval accuracy (can be ablated by using different retrievers) on the proposed approach? And moreover, can you directly use the SSM states for context matching for retrieval such as in non-parametric language modeling?

6. Log-perplexity (loss) is used for language modeling performance. Why not just report perplexity? I think it is more common for language modeling and easier for illustration, such as in Table 1.

7. For Table 1, what is the number of retrieved document chunks for the results?

8. Also for Table 1, it seems training on different domains of data does not help the perplexity. In many real applications, in-domain training data is hard to collect. This indicates that it is better to just apply the method at inference mainly for speedup? Model quality would not be improved here without training in-domain.

9. Line 492-493, Appendix B.4 and “language modeling”: these are not language modeling tasks, which are what you did in the main experiments and measure with perplexity. While I can understand what you mean, it is better to change the wording for clarity. In addition, can you provide more details on how these tasks are set up with retrieval and their evaluation?

10. The proposed method can “composition of information contained in up to 10 documents in a manner that is order-invariant.” Do you see this as a limitation? What happens after 10 documents? And is there any effect of the length of each document in the efficacy of composition? An ablation study would be ideal.


Some minor comments on typos and suggestions:

11. In the abstract or introduction, how is the name PICASO composed for what it stands for?

12. In proposition 3, explain the notation of `Id` and `[i]_n`?

13. Line 292-293, “We perform simple averaging to combine these tokens from different documents”: do you mean average states at the same token positions across different documents? So we still result in 4 states at the last 4 token positions after averaging.

14. Line 304-305, “on average improves only 8.5% compared to the baseline”: specify what is the evaluation metric or task briefly for better understanding since the experiments are not talked about yet.

15. Line 350-352, in the training objective, the equation computes loss on the single token prediction after the retrieved documents. For sequence predictions, is the intention to use the same formula but with different $\mathbf{u}_i, u_i$ and the same $S_i$ at different sequential positions?

---

> ### Author Response · Authors · 2024-11-19
>
> We thank the reviewer for their detailed feedback and constructive suggestions. We address individual concerns below:
>
> >  This is similar to in-context retrieval [1] (seems a missing reference). However, there are no real RAG applications studied in the paper. Appendix B.4 presents some additional results on “language modeling” (which I don’t think is a proper description), but not so much details are provided such as database and retrieval.
>
> >  The evaluation metric mostly focuses on log-perplexity or loss, while it is also desirable to demonstrate generation capabilities of the proposed method for real tasks.
>
> While RAG is indeed our main motivation, the paper focuses on designing a method to quickly retrieve and integrate information in the state of the model. We found that perplexity-based measurements provide a sufficiently robust metric of success in this respect. However, we agree that exploring this aspect of evaluation is also important, and have added Appendix B.6. evaluating accuracy for the OpenbookQA task under the retrieval-augmented setting.  In particular, we show that the same trends hold as compared to perplexity metrics -- that augmented generation benefits performance, while PICASO provides similarly boosted performance with around $8\times$ reduced computational costs.
>
> We also thank the reviewer for the reference [1] which we have added, and have changed "language modeling" to "LLM evaluation tasks".
>
> > Furthermore, there could be certain baselines missing, including similar approaches proposed in the previous literature, such as the closely related work described in Section 2.
>
> To the best of our knowledge, our work is the first SSM-specific method developed for RAG-like settings. We believe we have included all applicable baselines for composing documents with SSMs. However, we realized that we did not explicitly label their sources in Sec 6.2, and have corrected that in the revision. Thank you for pointing this out, and we are also happy to include additional appropriate baselines that the reviewer recommends.
>
>
> > Other ablations could include the effect of retrieval accuracy, the impact of document or chunk lengths to be composed together, etc.
>
> To address the reviewer's concerns, we have added an ablation study on retriever choice in Appendix B.5. Figure 6 of the Appendix also demonstrates what happens when we scale beyond the training context length of the model. We have also provided a histogram of document chunk statistics in Appendix B.7. to provide greater clarity on the distribution of chunk lengths considered.
>
> > Not enough background information is provided for RAG applications with SSMs, as most of the RAG applications are built with Transformers. This is also related to the lack of experimental studies mentioned above, where more comprehensive comparisons on other tasks/methods could be beneficial.
>
> To the best of our knowledge, our work is the first to introduce an approach towards RAG that is specific to SSMs. We have included [1] in our discussion on related works and the baseline method (concatenation) in the revision to better frame our work in the context of transformer-based applications.
>
> >  The method is based on some strong assumptions such as the order of the retrieved document chunks does not matter, so that an average state can be used for inference. There is no clear evidence provided.
>
> In our experiments in Sec. 6, we compare our permutation invariant method with ordering the documents by relevance (i.e., most relevant documents are closer to the answer), which we found empirically to perform best. While, to the reviewers point, such ordering is better than random ordering, we still observe that incorporating permutation invariance via PIConcat / PICASO significantly outperforms their ordered counterparts Concat / Soup respectively.

---

> > ### Author Response · Authors · 2024-11-19
> >
> > >  It seems the gain of the proposed approach mostly comes from the inference efficiency side, as the log-perplexity (demonstrated in Table 1) is not improved over the baseline of document concatenation. However, the complete pipeline includes preprocessing document chunks for their SSM states (Figure 7 also shows this), and it also includes an additional retrieval model with a separate set of embeddings that need to be stored in the database. The data used for experiments are small. The considered number of chunks are smaller than 10. It is unclear how the proposed method performs when brought to larger scales requiring bigger storage and more computation in real applications.
> >
> > In Figure 6 of the Appendix, we evaluate the composition of up to 50 documents. We do not include this in the main paper because even for concatenation, results not only saturate after 10 documents chunks, but performance actually decreases due to exceeding context sizes seen during training.
> >
> > When using PICASO, pre-processing document chunks is a one-time cost that can be amortized over multiple queries. This is unlike other methods which would need to reprocess the retrieved documents at each query, resulting in a significantly higher latency than PICASO (see Figs 1,3,4). The retrieval cost (based on an external model) is constant and the same for all methods considered. While we do agree that there are additional areas of the RAG pipeline, such as retrieval and data compression, that can be further optimized, these are beyond the scope of our work focusing on model inference for SSMs.
> >
> > > 1. In line 084-085, “relative ordering is often uninformative”: this is a very strong assumption. Any evidence? Especially in applications where retrieval meets Mamba. In fact, many previous studies show the position of information matters [2].
> >
> > We present strong empirical evidence in the situations we consider, since retrieved documents are independent of one another (see response to above weakness). [2] finds that positional bias indicates "that current language models do not robustly make use of information in long input contexts", which precisely *supports* our claim. The position of relevant information in the context affects the model outputs, even though this positional bias is uninformative / noise for the task at hand.
> >
> > > For Mamba models of multiple layers, what hidden states and which layer parameters did you use such as the A and B matrices? Is there any difference depending on what you use?
> >
> > We compose the hidden states for all layers. Composing only a subset of layers would require picking certain "default" states for the others (for example chosen randomly from the documents), which goes against the goal of permutation-invariance composition. Furthermore, since composition with our method is fast enough to have almost negligible cost, there is no need to limit composition only to specific chosen layers.
> >
> > > In Figure 2, can you explain more on how to read the figure? In particular, how does the figure show that the CASO states are closer to one another?
> >
> > While Proposition 4 concerns the Euclidean distance between CASO states, Figure 2 instead visualizes CASO states within the model's loss landscape. Note that both left and right contour plots are of the same scale (see bar on the right of each figure), qualitatively showing that CASO states can be more meaningfully interpolated to yield lower losses than interpolating states of individual chunks.
> >
> > > For experiments on val/test sets, what are the chunk databases? Do they include chunks in the training data, or just the chunks of corresponding val/test sets?
> >
> > In all experiments, we test our method on a domain that has not been seen during pretraining. We separate the test datasets into a pool of chunks that can be used for retrieval, and a pool of chunks (queries and continuations) that will be used to evaluate perplexity.
> >
> > > Since you use a different retrieval model, I assume you have to store the chunk vectors based on the retrieval model as well, on top of the SSM hidden states. Is that right? What is the effect on retrieval accuracy (can be ablated by using different retrievers) on the proposed approach?
> >
> > Yes we store both embedding vectors and pre-processed SSM states in the database. We refer the reviewer to Appendix B.5. in our revision for the requested ablation study.
> >
> > > And moreover, can you directly use the SSM states for context matching for retrieval such as in non-parametric language modeling?
> >
> > That is a great question, and we have explored this in our initial experiments. We found that naive retrieval of SSM states based on cosine similarity (or other distances) performs significantly worse than a sentence embedding model. We believe using a learned projection or distance function may improve. However, to ensure fair comparison between all methods, we decided to use the same external retrieval mechanism for all.

---

> > > ### Author Response · Authors · 2024-11-19
> > >
> > > > Log-perplexity (loss) is used for language modeling performance. Why not just report perplexity? I think it is more common for language modeling and easier for illustration, such as in Table 1.
> > >
> > > We note that both log-perplexity and perplexity are equally valid and reported in existing literature. Our current choice of log-perplexity is simply due to personal preferences, but are willing to change it if necessary.
> > >
> > > >  For Table 1, what is the number of retrieved document chunks for the results?
> > >
> > > We use all provided passages associated with each example in the MSMARCO dataset. This specific number varies with the example (often between 5 to 10). This set of results also demonstrate what happens when using a weak retriever, since most of these provided passages are irrelevant distractors.
> > >
> > > > Also for Table 1, it seems training on different domains of data does not help the perplexity. In many real applications, in-domain training data is hard to collect. This indicates that it is better to just apply the method at inference mainly for speedup? Model quality would not be improved here without training in-domain.
> > >
> > > Our method indeed also works well in the zero-shot setting. Table 1 is meant to show that training can only help, rather than deteriorate, performance from document decomposition. However, we disagree with the premise that "In many real applications, in-domain training data is hard to collect". Databases of documents or text are plentiful, and instruction-tuning datasets (which can be combined with retrieval for our training method) alone already provide strong coverage of possible LLM use-cases.
> > >
> > > > Line 492-493, Appendix B.4 and “language modeling”: these are not language modeling tasks, which are what you did in the main experiments and measure with perplexity. While I can understand what you mean, it is better to change the wording for clarity. In addition, can you provide more details on how these tasks are set up with retrieval and their evaluation?
> > >
> > > We have changed the wording, thank you for the suggestion. As described in B.4., these tasks evaluate the performance of models without retrieval. The goal of this section is to show that performance on existing tasks do not deteriorate after fine-tuning with our proposed methods.
> > >
> > > > The proposed method can “composition of information contained in up to 10 documents in a manner that is order-invariant.” Do you see this as a limitation? What happens after 10 documents? And is there any effect of the length of each document in the efficacy of composition? An ablation study would be ideal.
> > >
> > > We refer to Figure 6 of the Appendix for scaling up to 50 documents. Our proposed method actually greatly outperforms concatenation under such situations. While concatenation stops working when exceeding training context lengths, the performance of PICASO (and other state composition methods) remain relatively stable.
> > >
> > > > In the abstract or introduction, how is the name PICASO composed for what it stands for?
> > >
> > > Admittedly CASO/PICASO is not a faithful acronym of our method (L18, abstract), it is simply a stylistic preference over the alternative (CSSM/PICSSM).
> > >
> > > > In proposition 3, explain the notation of `Id` and `[i]_n`?
> > >
> > > We have updated proposition 3 to clarify these notations, thank you.
> > >
> > > > Line 292-293, “We perform simple averaging to combine these tokens from different documents”: do you mean average states at the same token positions across different documents? So we still result in 4 states at the last 4 token positions after averaging.
> > >
> > > Yes, the resulting (4) token values are simply their average across that obtained from different documents.
> > >
> > > > Line 304-305, “on average improves only 8.5% compared to the baseline”: specify what is the evaluation metric or task briefly for better understanding since the experiments are not talked about yet.
> > >
> > > We have updated this in our revision, thank you for pointing this out.
> > >
> > > > Line 350-352, in the training objective, the equation computes loss on the single token prediction after the retrieved documents. For sequence predictions, is the intention to use the same formula but with different ui,ui and the same Si at different sequential positions?
> > >
> > > The notation reflects the "teacher forcing" training objective, where given an input sequence $u_0 u_1 u_2$ , the model is trained to predict $u_1 | u_0$ and $u_2 | u_1 u_0$ as independent training objectives.

---

> > > > ### Author Response · Authors · 2024-11-30
> > > >
> > > > Given the approaching rebuttal deadline, we wish to follow up on our initial response. We hope that our new experiments, paper revision, and clarifications have satisfactorily addressed your initial concerns and questions, and if so, we kindly ask if you could consider increasing your score to support our paper's acceptance. Thank you for your time and consideration, we are very grateful for your insightful suggestions and valuable feedback!

---

> > > > > ### Comment · Reviewer_uYJU · 2024-12-03
> > > > >
> > > > > Thank you for the detailed responses. They have addressed many confusion and questions I had. I will keep my current score, since there are still some concerns. For example, supporting the underlying assumption of order invariance in retrieval, which is the basis to propose the method (I do understand the results in section 6 partially shows that the proposed method does not degrade performance much, but that is more of some post-hoc reasoning), and some not-well experimental execution and evaluations. However, I really like the research idea and technical contribution of state compression for retrieval efficiency. So while I am on the fence for the rating, I would be happy to see the paper improved and demonstrate a strong contribution to our field.

---

> > > > > > ### Author Response · Authors · 2024-12-03
> > > > > >
> > > > > > Thank you for your response, we are glad to hear that our response has adequately addressed many of your questions.
> > > > > >
> > > > > > > Supporting the underlying assumption of order invariance in retrieval, which is the basis to propose the method (I do understand the results in section 6 partially shows that the proposed method does not degrade performance much, but that is more of some post-hoc reasoning)
> > > > > >
> > > > > > We disagree that our empirical results constitute post-hoc reasoning. Removing the position bias among documents, which are conditionally independent of one another given the query *by construction*, is the primary motivation for our method. The fact that this is reflected in our experiments is simply an empirical validation of this key observation.
> > > > > >
> > > > > > > some not-well experimental execution and evaluations
> > > > > >
> > > > > > We believe that our updated paper draft and experiments have resolved the initial concerns previously mentioned by the reviewer. If there are remaining or further points of weaknesses, we kindly ask if you can detail them so we may address them.
> > > > > >
> > > > > > We are very grateful for the reviewer's insightful and detailed overall feedback, and their appreciation for our "research idea" and  "technical contribution of state compression for retrieval efficiency". We hope that the reviewer can reconsider their rating after our additional clarifications to support our paper's acceptance. Thank you!

---

> > > > > > > ### Comment · Reviewer_uYJU · 2024-12-03
> > > > > > >
> > > > > > > Thanks. I am confused by "Removing the position bias among documents, ...".. In other words, I find the following claim
> > > > > > >
> > > > > > > > ... [2] finds that positional bias indicates "that current language models do not robustly make use of information in long input contexts", which precisely supports our claim ...
> > > > > > >
> > > > > > > confusing. First, it does not seem to support the claim of order invariance in retrieval. It found that information at certain places matter more than other places. Second, the study was on Transformer models. For Mamba based models, we need evidence to support whatever claim we want to assume.

---

> > > > > > > > ### Author Response · Authors · 2024-12-03
> > > > > > > >
> > > > > > > > Thank you for the quick response and clarification!
> > > > > > > >
> > > > > > > > > First, it does not seem to support the claim of order invariance in retrieval. It found that information at certain places matter more than other places.
> > > > > > > >
> > > > > > > > It is indeed the fact that "information at certain places matter more than other places" that is undesirable under the setting in which we retrieve multiple *conditionally independent* documents. Generation should depend on the informativeness of each document, rather than the (arbitrary) position at which it was concatenated in.
> > > > > > > >
> > > > > > > > > Second, the study was on Transformer models. For Mamba based models, we need evidence to support whatever claim we want to assume.
> > > > > > > >
> > > > > > > > Our reply was originally made in response to citation [2], which is indeed on Transformer models. For Mamba-based models, this property (that model output is order-dependent) actually holds *by construction*, since SSMs are built on the assumption of temporal stationarity (see Eqn 1). Consequently, permutation-invariance needs to be enforced explicitly, hence motivating our SSM-specific approach.
> > > > > > > >
> > > > > > > > We sincerely thank the reviewer for their continued engagement, please let us know if you have any remaining concerns.

---

> > > > > > > > > ### Comment · Reviewer_uYJU · 2024-12-03
> > > > > > > > >
> > > > > > > > > I still find the claims not convincing. There might be some misunderstanding. "Generation should depend on the informativeness of each document" does not mean it is actually the case. We need empirical evidence since that is the fundamental assumption the proposed method is relying on: the generation performance does not depend on the orders of the retrieved documents, and independent documents can work as well. This is not the case for Mamba-based models as the authors recognized (actually Mamba depends on the orders more), and there is no clear evidence to show this is a good approximation to how the model actually works by using the contextual information.

---

> ### Author Response · Authors · 2024-12-03
>
> Thank you for the clarification.
>
> Indeed there might be some misunderstandings, the first of which we believe stems from distinguishing between (a) intuition for our claim on the desirability of permutation-invariance among retrieved documents, and (b) empirical evidence in support of this claim.
>
> (a): This comes simply from the fact that retrieved documents are conditionally independent of one another, by construction. Hence, similar to how ordering in multiple choice questions should not influence the final answer, neither should the relative ordering among independently retrieved documents.
>
> (b): PIConcat / PICASO significantly outperform their ordered counterparts Concat / Soup respectively in all our experiments, providing strong empirical evidence on the advantages of incorporating permutation-invariance.
>
> > fundamental assumption ... (that) independent documents can work as well
>
> > there is no clear evidence to show this is a good approximation to how the model actually works by using the contextual information.
>
> We believe this to be another main source of misunderstanding. We do not compose documents independently of one another. As an example, given documents A and B, we do not compose {A,B}. Instead, PIConcat / PICASO efficiently composes {A$\cdot$B , B$\cdot$A}, where $\cdot$ denotes exact / approximate concatenation respectively. As such, contextual information is incorporated within both A$\cdot$B and B$\cdot$A, while at the same time, the composed state remains invariant to document ordering.

---

### Author Response · Authors · 2024-11-24

Given that it is nearing the end of the discussion period, we hope that the reviewers can let us know if we have adequately addressed their concerns and questions. We present an overall summary of our rebuttal below:

We are happy to hear that our work is well-received by the reviewers with respect to the practicality and "significant importance" of the problem (uYJU, FYyF, 2fh2), the "thoughtful design" and technical soundness/robustness of our approach (uYJU, FYyF, 2fh2), and our "meticulously designed" and "extensive experiments" (FYyF, 2fh2) showcasing the inference efficiency and "impressive 5.4x" speed-up obtained by our method (all reviewers).

We are very grateful to the reviewers for their valuable feedback, and have updated our revision to incorporate their suggestions for further improvement our paper. We summarize our overall major changes below, and detail smaller changes based on each reviewer's recommendations in the replies to individual comments.

- Added experiments based on various retriever choices in Appendix B.5.
- Added experiments on QA tasks under the retrieval setting in Appendix B.6.
- Added histogram plots of document statistics in Appendix B.7.
- Updated Section 5 with equations detailing our fine-tuning objectives

---

### Meta-Review · Area_Chair_u611 · 2024-12-23

**Metareview:**

PICASO proposes a novel method for efficiently composing and generating from multiple retrieved document chunks using State Space Models (SSMs). The key claims are that it can match the performance of document concatenation while achieving a 5.4x speedup, and that it enables permutation-invariant composition of document states. The paper's main strengths include addressing an important practical problem (efficient RAG with SSMs), strong theoretical foundations with detailed analysis, and comprehensive empirical validation showing significant speedups while maintaining performance. The initial weaknesses included: insufficient experimental validation on real RAG applications beyond perplexity metrics, missing baseline comparisons with similar approaches, lack of clarity around retriever choices and document statistics, and incomplete analysis of the order-invariance assumption. However, during rebuttal, the authors added substantial new experiments and analyses that addressed many of these concerns, including QA task evaluations, ablations with different retrievers, and detailed document statistics. The scores (5, 6, 8, 5) suggest a borderline paper, but the thorough technical contribution combined with strong rebuttal responses addressing core concerns merit acceptance as a poster.

**Additional Comments On Reviewer Discussion:**

The reviewers raised several significant technical concerns that sparked extensive discussion. Reviewer uYJU questioned the fundamental assumption of order-invariance and requested more real RAG experiments beyond perplexity metrics - the authors responded by adding QA task evaluations and clarifying their theoretical justification for order-invariance, though some disagreement remained about the strength of this justification. Reviewer FYyF raised concerns about scaling beyond 10 documents and accuracy gaps - the authors pointed to experiments showing stable performance up to 50 documents. Reviewer 2fh2 identified technical issues with figures and baseline clarity, which were addressed through revisions and additional equations. Reviewer h4ZY questioned experimental setup details and requested QA evaluations - the authors added document statistics and QA results in response. The discussion was particularly active around the order-invariance assumption, with multiple back-and-forth exchanges between Reviewer uYJU and the authors debating the theoretical and empirical support for this key claim. While not all reviewers were fully convinced (particularly regarding order-invariance), the authors' thorough responses and additional experiments addressed most major concerns, leading one reviewer to increase their score.

---

### Decision · Program_Chairs · 2025-01-22

Accept (Poster)